# Individual and community-level determinants, and spatial distribution of institutional delivery in Ethiopia, 2016: Spatial and multilevel analysis

Getayeneh Antehunegn Tesema[1]*, Tesfaye Hambisa Mekonnen[2], Achamyeleh Birhanu Teshale[1]

1 Department of Epidemiology and Biostatistics, Institute of Public Health, College of Medicine and Health Sciences, University of Gondar, Gondar, Ethiopia, 2 Department of Environmental and Occupational Health and Safety, Institute of Public Health, College of Medicine and Health Sciences, University of Gondar, Gondar, Ethiopia

* getayenehantehunegn@gmail.com

## Abstract

### Background

Institutional delivery is an important indicator in monitoring the progress towards Sustainable Development Goal 3.1 to reduce the global maternal mortality ratio to less than 70 per 100,000 live births. Despite the international focus on reducing maternal mortality, progress has been low, particularly in Sub-Saharan Africa (SSA), with more than 295,000 mothers still dying during pregnancy and childbirth every year. Institutional delivery has been varied across and within the country. Therefore, this study aimed to investigate the individual and community level determinants, and spatial distribution of institutional delivery in Ethiopia.

### Methods

A secondary data analysis was done based on the 2016 Ethiopian Demographic and Health Survey (EDHS) data. A total weighted sample of 11,022 women was included in this study. For spatial analysis, ArcGIS version 10.6 statistical software was used to explore the spatial distribution of institutional delivery, and SaTScan version 9.6 software was used to identify significant hotspot areas of institutional delivery. For the determinants, a multilevel binary logistic regression analysis was fitted to take to account the hierarchical nature of EDHS data. The Intra-class Correlation Coefficient (ICC), Median Odds Ratio (MOR), Proportional Change in Variance (PCV), and deviance (-2LL) were used for model comparison and for checking model fitness. Variables with p-values<0.2 in the bi-variable analysis were fitted in the multivariable multilevel model. Adjusted Odds Ratio (AOR) with a 95% Confidence Interval (CI) were used to declare significant determinant of institutional delivery.

### Results

The spatial analysis showed that the spatial distribution of institutional delivery was significantly varied across the country [global Moran's I = 0.04 (p<0.05)]. The SaTScan analysis

**Funding:** The author(s) received no specific funding for this work.

**Competing interests:** The authors have declared that no competing interests exist.

**Abbreviations:** ANC, Antenatal Care; AOR, Adjusted Odds Ratio; ARR, Annual Rate of Reduction; BMI, Body Mass Index; CI, Confidence Interval; DHS, Demographic Health Survey; EA, Enumeration Area; EDHS, Ethiopian Demographic Health Survey; GIS, Geographic Information System; ICC, Intra-cluster Correlation Coefficient; LLR, log-likelihood Ratio; LR, Likelihood Ratio; MOR, Median Odds Ratio; PCV, Proportional Change in Variance; WHO, World Health Organization.

identified significant hotspot areas of poor institutional delivery in Harari, south Oromia and most parts of Somali regions. In the multivariable multilevel analysis; having 2–4 births (AOR = 0.48; 95% CI: 0.34–0.68) and >4 births (AOR = 0.48; 95% CI: 0.32–0.74), preceding birth interval $\geq$ 48 months (AOR = 1.51; 95% CI: 1.03–2.20), being poorer (AOR = 1.59; 95% CI: 1.10–2.30) and richest wealth status (AOR = 2.44; 95% CI: 1.54–3.87), having primary education (AOR = 1.47; 95% CI: 1.16–1.87), secondary and higher education (AOR = 3.44; 95% CI: 2.19–5.42), having 1–3 ANC visits (AOR = 3.88; 95% CI: 2.77–5.43) and >4 ANC visits (AOR = 6.53; 95% CI: 4.69–9.10) were significant individual-level determinants of institutional delivery while being living in Addis Ababa city (AOR = 3.13; 95% CI: 1.77–5.55), higher community media exposure (AOR = 2.01; 95% CI: 1.44–2.79) and being living in urban area (AOR = 4.70; 95% CI: 2.70–8.01) were significant community-level determinants of institutional delivery.

## Conclusions

Institutional delivery was low in Ethiopia. The spatial distribution of institutional delivery was significantly varied across the country. Residence, region, maternal education, wealth status, ANC visit, preceding birth interval, and community media exposure were found to be significant determinants of institutional delivery. Therefore, public health interventions should be designed in the hotspot areas where institutional delivery was low to reduce maternal and newborn mortality by enhancing maternal education, ANC visit, and community media exposure.

## Background

Despite improvements over the last two decades, maternal mortality in developing countries, especially in sub-Saharan Africa (SSA), remains a significant public health concern [1, 2]. Globally, as a result of preventable causes of pregnancy and childbirth, about 358,000 maternal deaths occur annually, of which 99% occur in developing countries [3]. In high-income countries, maternal mortality has been decreased dramatically [4], but SSA continued to account for 66% of maternal deaths worldwide [5]. In SSA, like Ethiopia, pregnancy and childbirth-related complications such as Postpartum Hemorrhage (PPH), pregnancy-induced high blood pressure, fetal asphyxia, stillbirth, sepsis, obstructed labor, and unsafe abortions are unacceptably high, leading to the massive burden of maternal mortality [6–8].

The World Health Organization (WHO) recommends health facility delivery as a key strategy for reducing maternal and neonatal mortality [9]. Institutional delivery grants safe birth outcomes through the provisions of supportive facilities, clean delivery services with skilled experts, and early detection and management of maternal and neonatal complications [10]. Although institutional delivery are a key strategy for reducing pregnancy and birth risks, many women in developing countries give birth at home [11]. For example, the prevalence of institutional delivery in Asia and SSA is lower than 50% [12]. Thus, it varies from 26% in Ethiopia [13] to 67.3% in Tanzania [1].

According to prior studies conducted in Ethiopia, the prevalence of institutional delivery varied across the country. Studies conducted on the prevalence and associated factors of institutional delivery in different regions of Ethiopia showed that 18.2% of the mothers in the Oromia region [14], 4.1% in the Tigray region [15], 78.8% in the Amhara region [16], 62.2% in

southern Ethiopia [17], 31% in the Gurage zone [18], 14.4% in West Shewa [19], 18.3% in Northwest Ethiopia [20] gave birth at the health facility.

Previous literature revealed that sex of household head, maternal age, maternal occupation, parity, birth order [21–25], number of Antenatal Care (ANC) visits [1, 12, 26–28], knowledge towards danger signs of pregnancy and childbirth [1, 21, 22], household wealth index [1, 25, 29], media exposure [30], maternal and parental education [1, 26, 29], previous history of prolonged labour [31], number of children [31, 32], birth preparedness/complication readiness [32, 33], and decision making on health care [8, 9, 12, 16] were the individual-level predictors significantly associated with institutional delivery. Studies also documented that community-level factors such as region, residence [25, 26, 32], distance to the nearest health facility, and community media exposure [2, 34] were significantly associated with institutional delivery.

Despite Ethiopia having made a large scale investment to reduce maternal and neonatal mortality through free of charge maternal health care services such as ANC, institutional delivery, and PNC [35, 36], still maternal and newborn mortality is highest in Ethiopia [9]. It is common in rural parts of the county, where it is more challenging to get access to health facilities, and home delivery is highly practiced [13].

In Ethiopia, previous studies were done on the prevalence and associated factors of institutional delivery [28, 30, 37] and reported that the prevalence had been varied across the country [15–19, 38] but none of these studies have tried to explore the spatial distribution of institutional delivery in Ethiopia. Besides, there are two studies on institutional delivery based on the nationally representative Ethiopian Demographic and Health Survey (EDHS) data [28, 30]. These studies were failed to capture the spatial distribution of institutional delivery in Ethiopia, and the data they used were not weighted data. Therefore, we aimed to investigate the individual and community-level determinants, and spatial distribution of institutional delivery in Ethiopia based on weighted 2016 EDHS data. Thus, the identifications of significant hotspot areas with a low prevalence of institutional delivery have become indispensable to design targeted effective public health interventions to enhance institutional delivery and reduce maternal and newborn mortality in Ethiopia. Furthermore, this study's findings could guide policymakers to work on individual and community-level determinants to improve institutional delivery in the country to strengthen maternal and child health.

## Methods

### Study design, setting, and period

A secondary data analysis was done based on the 2016 EDHS data. The EDHS was a nationally representative study conducted every five years in Ethiopia. Ethiopia is situated in the Horn of Africa. It has 9 Regional states (Afar, Amhara, Benishangul-Gumuz, Gambela, Harari, Oromia, Somali, Southern Nations, Nationalities, and People's Region (SNNP) and Tigray regions) and two city Administrations (Addis Ababa and Dire-Dawa) (Fig 1). About 84% of the population lives in rural areas [39]. In EDHS 2016, a two-stage stratified cluster sampling technique was employed using the 2007 Population and Housing Census (PHC) as a sampling frame. In the first stage, 645 EAs (202 in the urban area) were selected, and in the second stage, on average 28 households were systematically selected. A total of 18,008 households and 16,583 eligible women were included. The detailed sampling procedure was presented in the full EDHS 2016 report [40]. The source population was all women of reproductive age who gave birth in Ethiopia within five years before the survey, while the sample population was all women of reproductive age who gave birth in the selected EAs within five years before the survey. A total weighted sample of 11,022 reproductive-age women who gave birth within five years preceding the survey was included in this study.

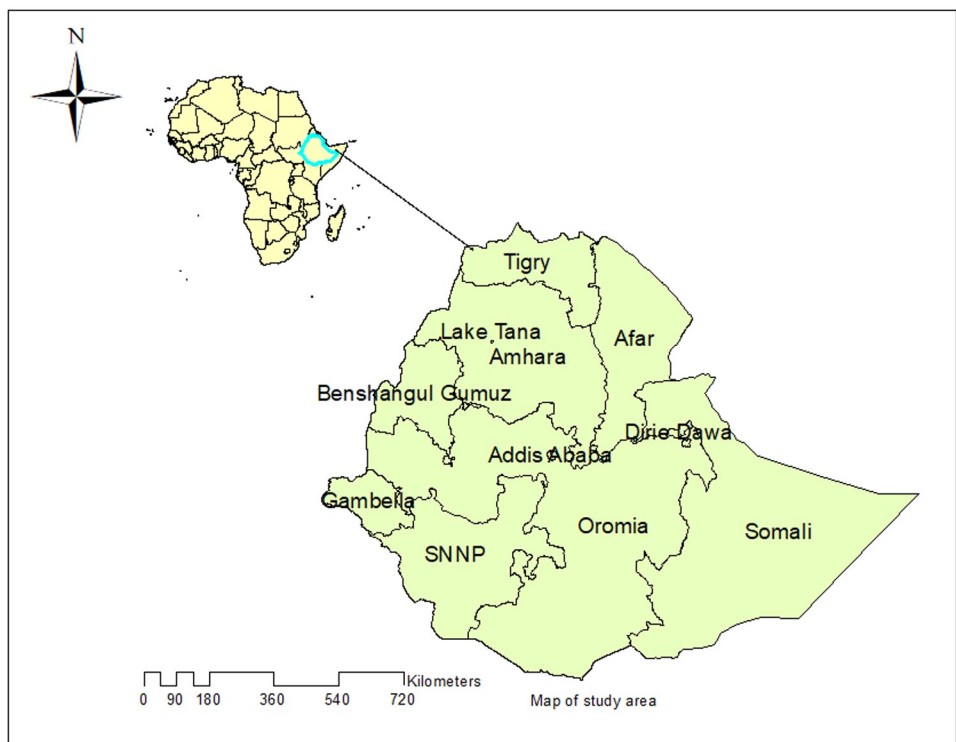

**Fig 1. Map of the study area (Source, CSA: 2013).**

## Study variables

**Outcome variable.** The dependent variable was whether a woman who gave birth within five years preceding the survey was delivered at a health facility or at home. We used the "place of delivery" as the outcome variable and recoded as home delivery (when the birth took place at home) and institutional delivery (when the birth took at the hospital, health center, or health post).

**Independent variables.** Consistent with the study's objective and given the hierarchical structure of EDHS data where women were nested within the cluster, two levels of independent variables were considered. At level-1 contained individual-level variables such as age, maternal education, husband education, media exposure, wealth index, sex of household head, ANC visit, parity, preceding birth interval, multiple gestations, religion, ever had of a terminated pregnancy, and birth order was included. At level-2 the community-level variables considered in this study were region, residence, community media exposure, and distance to get health facility.

In EDHS data, there was no variable collected at the community level except region (recoded as pastoralist region (Benishangul, Somali, Gambella, and Afar), Semi-pastoralist (Oromia, SNNPR), Agrarian (Amhara and Tigray) and City administration (Addis Ababa, Dire Dawa, and Harari)), distance to get health facility (recorded as a big problem and not a big problem), and residence (recoded as urban and rural). Therefore, we generated community media exposure by aggregating listening radio, watching television, and reading newspapers at the cluster level. These were categorized as higher community media exposure and lower media exposure based on the national median value of media exposure since it was not normally distributed [41].

## Data management and analysis

The data were weighted using sampling weight, primary sampling unit, and strata before any statistical analysis to restore the representativeness of the survey and to tell the STATA to take into account the sampling design when calculating standard errors, to get reliable statistical estimates. Descriptive and summary statistics were conducted using STATA version 14 software.

**Spatial analysis.** *Spatial autocorrelation analysis.* ArcGIS version 10.6 software was used to explore the spatial distribution of institutional delivery. The global spatial autocorrelation (Global Moran's I) was done to assess whether institutional delivery patterns were dispersed, clustered, or randomly distributed in the study area [42]. Moran's I is a spatial statistic used to measure spatial autocorrelation by taking the entire data set and producing a single output value ranging from -1 to +1. Moran's I value close to −1 indicates the spatial distribution of institutional delivery is dispersed, whereas Moran's I close to +1 indicate spatial distribution of institutional delivery is clustered. The Moran I value close to 0 means the spatial distribution of institutional delivery is random. A statistically significant Moran's I ($p < 0.05$) indicates the spatial clustering of institutional delivery.

*Spatial interpolation.* The spatial interpolation was done to predict institutional delivery on the un-sampled areas in the country based on sampled measurements. Ordinary Kriging (OK) and Empirical Bayesian Kriging (EBK) were done since it statistically optimizes the weight [43], to predict the prevalence of institutional delivery on the unobserved areas based on the observed measurement. The ordinary Kriging spatial interpolation method was selected for this study for predictions of institutional delivery since it had a smaller residual and Root Mean Square Error (RMSE) than EBK.

*Spatial scan statistical analysis.* In the spatial scan statistical analysis, Bernoulli based model was employed to identify statistically significant spatial clusters of institutional delivery using Kuldorff's SaTScan version 9.6 statistical software. For this study, we used a circular scanning window that moves across the study area since the elliptical window is inactive in the SaTScan software. Women with home delivery were taken as cases and those who had institutional delivery were considered as controls to fit the Bernoulli model. The numbers of cases in each location had Bernoulli distribution and the model required data for cases, controls, and geographic coordinates. The default maximum spatial cluster size of <50% of the population was used as an upper limit, which allowed both small and large clusters to be detected and ignored clusters that contained more than the maximum limit.

For each potential cluster, a likelihood ratio test statistic and the p-value were used to determine if the number of observed home delivery within the potential cluster was significantly higher than expected or not. The scanning window with maximum likelihood was the most likely performing cluster, and the p-value was assigned to each cluster using Monte Carlo hypothesis testing by comparing the rank of the maximum likelihood from the real data with the maximum likelihood from the random datasets. The primary and secondary clusters were identified and assigned p-values and ranked based on their likelihood ratio test based on 999 Monte Carlo replications [44].

**Multilevel analysis.** There is a hierarchical nature of the EDHS data; therefore, women have been nested within a cluster, and we assume that women in the same cluster may share similar characteristics to women in another cluster. These violate the usual hypothesis of the logistic regression model, which is the independence of observations and equal variance between clusters. This implies the need to take into account the heterogeneity between clusters by using an advanced model. Therefore, a multilevel binary logistic regression model was performed.

The i[th] mother's response variable is represented by a random variable Yi with two possible values coded as 1 and 0. So, the i[th] mother Yi's response variable was measured as a dichotomous variable with possible values Yi = 1, if ith mother gave birth in the institution and Yi = 0 if a mother gave birth in their home. We will fit the multilevel model by

$$\text{Log} \left[ \pi ij / \left( 1 - \pi ij \right) \right] \, = \beta 0 + \beta 1 xij + \beta 2 xij \ldots . + u0j + e0ij$$

Where:
$\Pi ij$: the probability of having institutional delivery
$1 - \pi ij$: the probability of having home delivery
$\beta 0$: the intercept
$\beta 1 / Bn$: regression coefficient of individual and community level factors
$u0j$: random errors at cluster levels
$e0ij$: random error at the individual level

Model comparison was made based on deviance (-2LL) since the models were nested models, and a model with the lowest deviance was the best-fitted model for the data. Likelihood Ratio (LR) test, Intra-class Correlation Coefficient (ICC), Median Odds Ratio (MOR), and Proportional Change in Variance (PCV) were computed to measure the variation of institutional delivery between clusters. The ICC quantifies the degree of heterogeneity of institutional delivery between clusters (the proportion of the total observed variation in institutional delivery that is attributable to between cluster variations).

ICC = $\sigma^2 / (\sigma^2 + \pi^2 / 3)$ [45], but MOR quantifies the variation or heterogeneity in institutional delivery between clusters in terms of odds ratio scale and is defined as the median value of the odds ratio between the cluster at high likelihood of institutional delivery and cluster at lower risk when randomly picking out individuals from two clusters (EAs).

$$\text{MOR} \, = \, exp^{\sqrt{(2*\partial 2*0.6745)}}, \, \text{MOR} \, = \, exp^{0.95*\partial}$$

[46].

$\partial^2$ indicates that cluster-level variance

PCV measures the total variation attributed to the final multilevel model as compared to the null model. We calculated the percentage of the Proportional Change in Variance (PCV) as follows

$$\text{PCV} = \frac{\text{var (null model)} - \text{var (final model)}}{\text{Var (null model)}}$$

Where; var (null model) = variance of the initial model, and var (final model) = variance of the final model. PCV measures the variation in institutional delivery explained by the full model (a model with both individual and community level variables simultaneously). Total variance was calculated by adding individual level variance ($\pi^2 / 3$) and community level variance, as individual level variable binary model is $\pi^2 / 3$ (3.29).

A two-level multilevel binary logistic regression model was used to analyze factors associated with institutional delivery. Four models were constructed for the multilevel logistic regression analysis. The first model was a null model without explanatory variables to determine the extent of cluster variation in institutional delivery. The second model was adjusted with individual-level variables; the third model was adjusted for community-level variables while the fourth was fitted with both individual and community level variables simultaneously.

Variables with p-value <0.2 in the bi-variable analysis for both individual and community-level factors were fitted in the multivariable model. We used 0.2 because incorporating variables with p-value up to 0.2 is important since these variables might have a good contribution

in the multivariable analysis. Adjusted Odds Ratio (AOR) with a 95% Confidence Interval (CI) in the multivariable model were used to declare statistically significant determinants of institutional delivery. Multi-collinearity was also checked using the variance inflation factor (VIF) by doing pseudo linear regression analysis and indicates that there was no multi-collinearity since all variables have VIF <5 and tolerance greater than 0.1.

## Ethical consideration

Since the study was a secondary data analysis of publicly available survey data from the MEASURE DHS program, ethical approval and participant consent were not necessary for this particular study. We requested DHS Program and permission was granted to download and use the data for this study from http://www.dhsprogram.com. There were no names of individuals or household addresses in the data file. The geographic identifiers only go down to the regional level (where regions are typically very large geographical areas encompassing several states/provinces). Each enumeration area (Primary Sampling Unit) has a PSU number in the data file, but the PSU numbers do not have any labels to indicate their names or locations. In surveys that collect GIS coordinates in the field, the coordinates are only for the enumeration area (EA) as a whole, not for individual households. The measured coordinates are randomly displaced within a large geographic area so that specific enumeration areas cannot be identified.

## Results

### Socio-demographic and economic characteristics of participants

A total of 11,022 reproductive-age women who gave birth within five years preceding the survey were included in this study. Of these, 4,851 (44.0%) were from Oromia region and 26 (0.25%) were from Harari region. About 9,807 (89.0%) of the women were living in rural areas, and the majority (41.4%) of the respondents were Muslim followers. Nearly two-thirds (66.1%) of the women and a half (47.8%) of their husbands didn't have formal education. Regarding the age of the women, 7,910 (71.8%) were in the age group of 20–34 years (Table 1).

### Obstetric and maternal service-related characteristics of respondents

Nearly half (43.9%) of women had 2–4 births, and about 91.2% had no prior pregnancy termination history. Of the total, 4,738 (43.0%) of women had a preceding birth interval of 24 to 48 months, and the majority (60.6%) of respondents reported as the distance to reach a health facility was a big problem (Table 2).

### Regional prevalence of institutional delivery in Ethiopia, 2016

The overall prevalence of institutional delivery in Ethiopia was 26.2% [95 CI: 25.4%, 27.1%], which was significantly varied across regions ranging from 14.7% in the Afar region to 96.6% in Addis Ababa (Fig 2).

### Spatial analysis

**Spatial autocorrelation analysis.** The global spatial autocorrelation analysis revealed that the spatial distribution of institutional delivery was significantly varied across the country with Global Moran's Index value of 0.04 (p<0.05) (Fig 3). In this study, areas with a low prevalence of institutional delivery were identified in Addis Ababa, Dire-Dawa, and Tigray regions. In contrast, areas with a high prevalence of institutional delivery were detected in Amhara, Afar, Somali, and Gambella regions (Fig 4).

**Table 1. Socio-demographic and economic characteristics of respondents in Ethiopia, 2016.**

| Variable | Frequency (N = 11,022) | Percentage |
|---|---|---|
| **Region** | | |
| Tigray | 716 | 6.5 |
| Afar | 114 | 1.0 |
| Amhara | 2,072 | 18.8 |
| Oromia | 4,851 | 44.0 |
| Somali | 508 | 4.6 |
| Benishangul | 122 | 1.1 |
| SNNPs | 2,296 | 20.8 |
| Gambella | 27 | 0.2 |
| Harari | 26 | 0.2 |
| Addis Ababa | 244 | 2.2 |
| Dire Dawa | 47 | 0.4 |
| **Residence** | | |
| Urban | 1,215 | 11.0 |
| Rural | 9,807 | 89.0 |
| **Religion** | | |
| Orthodox | 3,772 | 34.2 |
| Muslim | 4,561 | 41.4 |
| Catholic | 103 | 0.9 |
| Protestant | 2,329 | 21.1 |
| Traditional | 257 | 2.3 |
| **Maternal age (in years)** | | |
| < 20 | 378 | 3.4 |
| 20–34 | 7,910 | 71.8 |
| ≥ 35 | 2,734 | 24.8 |
| **Maternal education** | | |
| No education | 7,284 | 66.1 |
| Primary education | 2,951 | 26.8 |
| Secondary education | 514 | 4.7 |
| Higher education | 274 | 2.5 |
| **Husband education** | | |
| No education | 5,003 | 47.8 |
| Primary | 4,115 | 39.3 |
| Secondary | 797 | 7.6 |
| Higher | 544 | 5.3 |
| **Wealth status** | | |
| Poorest | 2,636 | 23.9 |
| Poorer | 2,520 | 22.9 |
| Middle | 2,280 | 20.7 |
| Rich | 1,998 | 18.1 |
| Richest | 1,588 | 14.4 |
| **Media exposure** | | |
| No | 7,375 | 66.9 |
| Yes | 3,647 | 33.1 |
| **Sex of household head** | | |
| Male | 9,494 | 86.1 |
| Female | 1,528 | 13.9 |

**Table 2. Obstetric and maternal service-related characteristics of respondents in Ethiopia, 2016.**

| Variable | Frequency | Percentage |
|---|---|---|
| **Parity** | | |
| 1 | 1,434 | 13.0 |
| 2–4 | 4,836 | 43.9 |
| 5⁺ | 4,752 | 43.1 |
| **Multiple gestation** | | |
| No | 10,730 | 97.4 |
| Yes | 292 | 2.6 |
| **Preceding birth interval** | | |
| < 24 month | 1,942 | 17.6 |
| 24–48 month | 4,738 | 43.0 |
| >48 month | 4,343 | 39.4 |
| **Distance to health facility** | | |
| Big problem | 6,676 | 60.6 |
| Not a big problem | 4,346 | 39.4 |
| **Ever had of a terminated pregnancy** | | |
| No | 10,056 | 91.2 |
| Yes | 966 | 8.8 |
| **Birth order** | | |
| 1 | 2,058 | 18.7 |
| 2 | 1,784 | 16.2 |
| ≥3 | 7,180 | 65.1 |
| **Number of ANC visit** | | |
| None | 2,818 | 37.1 |
| 1–3 visit | 2,342 | 30.9 |
| ≥ 4 visits | 2,429 | 32.0 |

**Spatial interpolation.** In the Kriging interpolation analysis, the highest prevalence of institutional delivery was detected in Addis Ababa, Dire Dawa, Harari, central Gambella, and Tigray regions. In contrast, the predicted low prevalence of institutional delivery was identified in Afar, east Somali, southwest Oromia, Benishangul, and Amhara regions (Fig 5).

**Spatial scan statistical analysis.** A spatial scan statistical analysis identified a total of 331 significant clusters, of which 104 were most likely (primary) clusters, and 227 were secondary clusters. The primary clusters were located in Harari, south Oromia, and most parts of Somali regions centered at 4.180558 N, 42.052871 E with 567.56 km radius, a Relative Risk (RR) of 1.24 and Log-Likelihood Ratio (LLR) of 106.5, at p < 0.0001 (Fig 6, Table 3). It showed that women inside the spatial window had a 1.24 times higher likelihood of having home delivery than women outside the spatial window.

## Individual and community-level determinants of institutional delivery

**The random effect analysis result.** In the null model, the ICC indicated that 57% of the total variability for institutional delivery was due to differences between clusters while the remaining unexplained 43% of the total variability of institutional delivery was attributable to the individual differences. Moreover, the MOR was 7.01 (95% CI: 6.02, 9.07) in the null model, which indicated that there was variation in institutional delivery between clusters. If we randomly select two women from different clusters, if we transfer women from low institutional delivery clusters to higher institutional delivery clusters, she could have 7.01 times higher odds

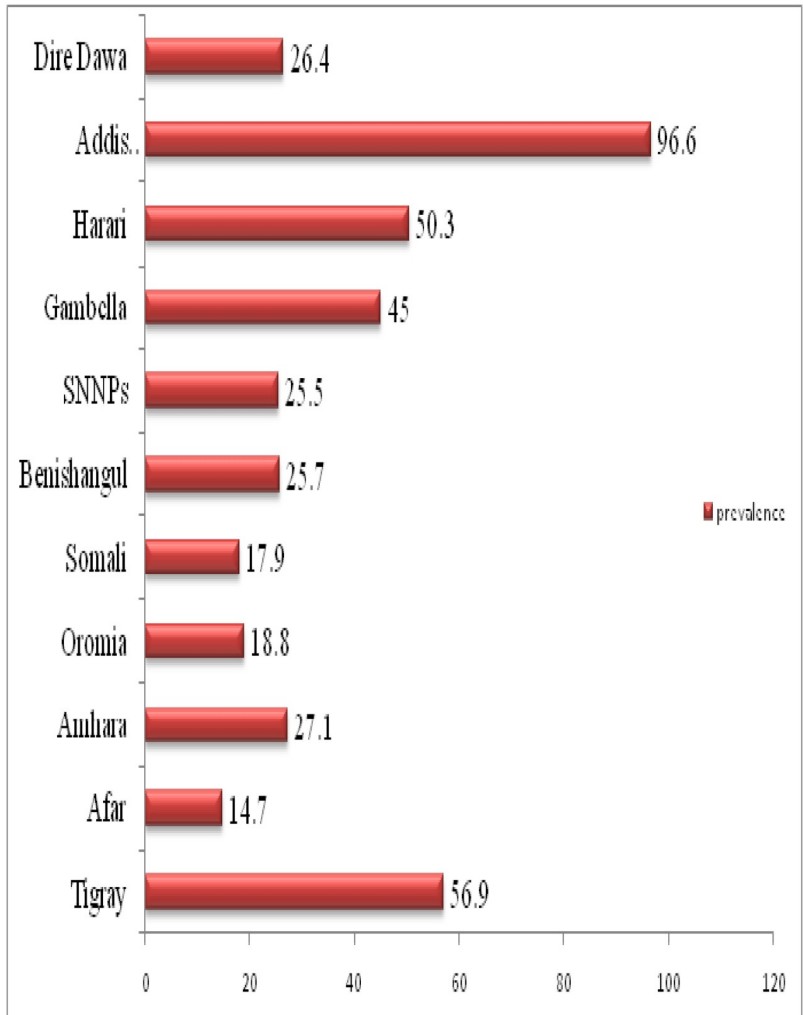

**Fig 2. Regional prevalence of institutional delivery in Ethiopia, 2016.**

of having institutional delivery. The PCV in the final model was 73%, it showed that about 73% of the variability in institutional delivery was explained by the full model (a model with individual and community level variables). Deviance was used to compare the fitted models and model 3 with the lowest deviance value was the best-fitted model (Table 4).

**The fixed effect analysis result.** In the multivariable multilevel logistic regression analysis parity, preceding birth interval, the number of ANC visits, wealth status, residence, community media exposure, region, and maternal education were significantly associated with institutional delivery.

The odds of having institutional delivery among women who had 2–4 births and more than four births were decreased by 62% (AOR = 0.48; 95% CI: 0.34–0.68) and 62% (AOR = 048; 95% CI: 032–0.74) as compared to primiparous women respectively. Women who had preceding birth interval ≥ 48 months had 1.51 (AOR = 1.51; 95% CI: 1.03–2.20) times higher odds of giving birth at health institutions compared to women who had preceding birth interval less than 24 months. The odds of having institutional delivery for women who had primary education, and secondary and above education were 1.47 (AOR = 1.47; 95% CI: 1.16–1.87) and 3.44

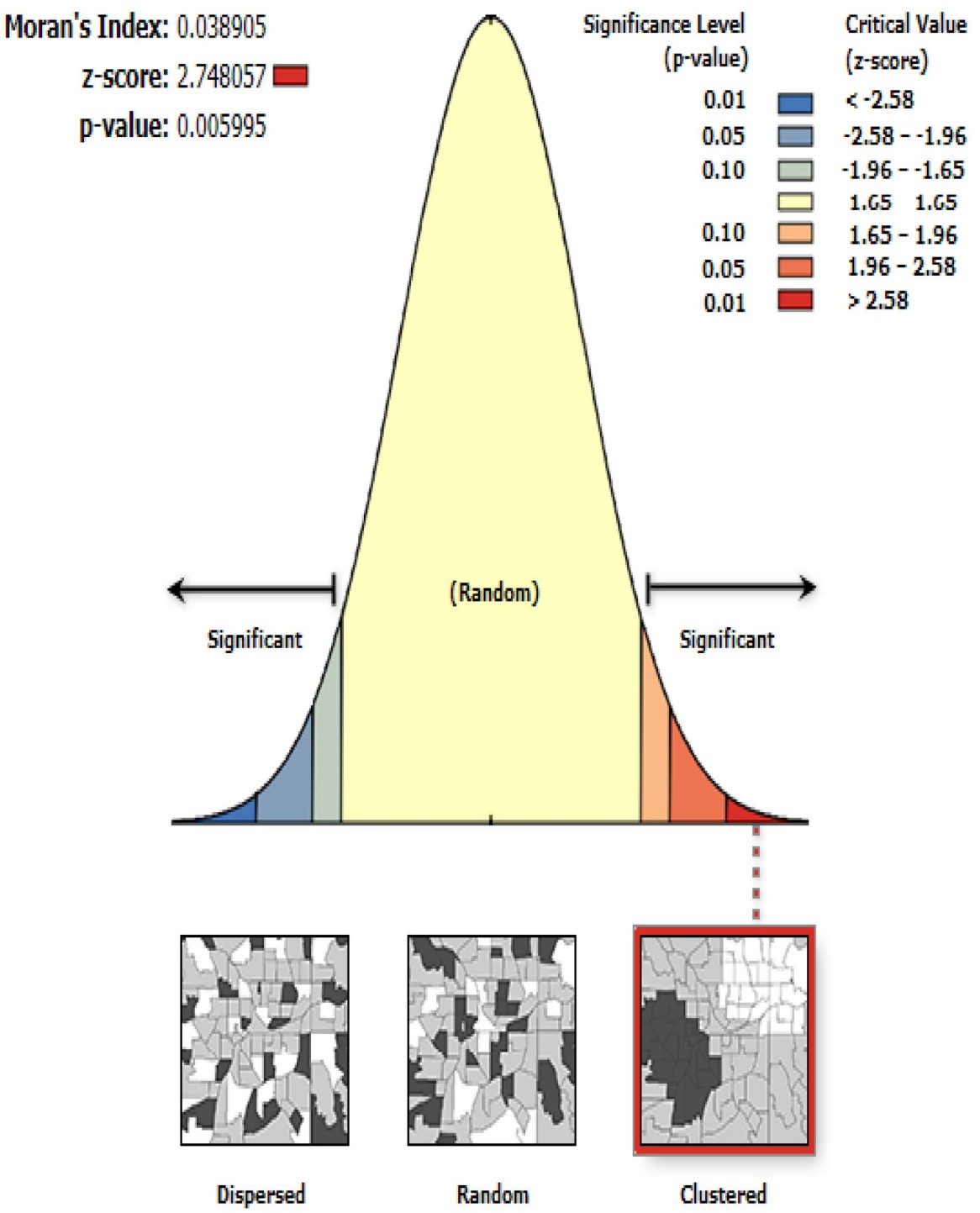

**Fig 3. Global autocorrelation of institutional delivery in Ethiopia, 2016.**

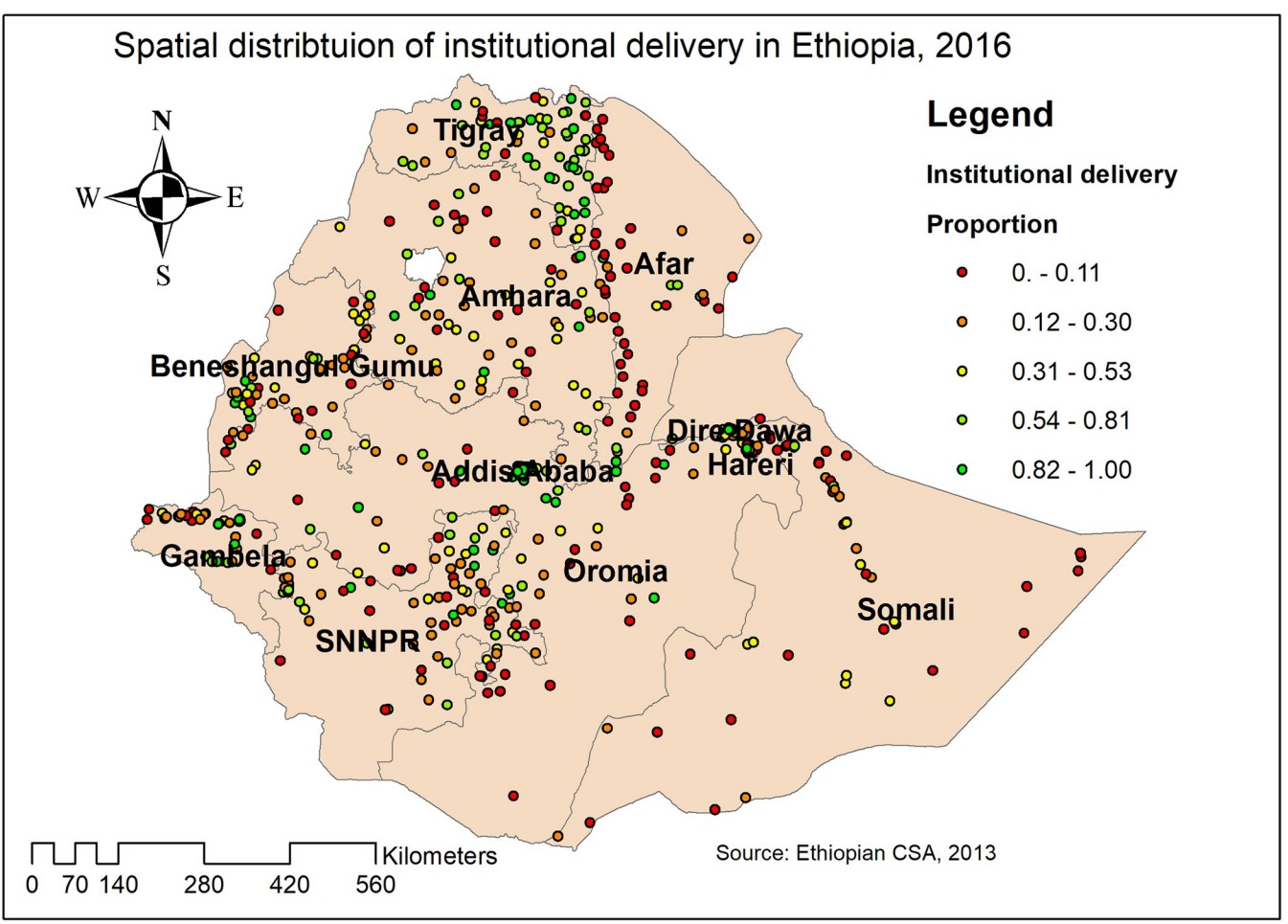

**Fig 4. Spatial distribution of institutional delivery in Ethiopia, 2016 (Source, CSA: 2013).**

(AOR = 3.44; 95% CI: 2.19–5.42) times more likely to have institutional delivery than women who had no formal education respectively.

Women in the poor and richest households had 1.57 (AOR = 1.57; 95% CI: 1.10–2.30) and 2.44 (AOR = 2.44; 95% CI: 1.54–3.87) times higher odds of having institutional delivery than women in the poorest household, respectively. Mother who had 1–3 ANC visit and ≥ 4 ANC visit for the index pregnancy was 3.88 (AOR = 3.88; 95% CI: 2.77–5.43) and 6.53 (AOR = 6.53; 95% CI: 4.69–9.10) times higher odds of having institutional delivery as compared to mother who had no ANC visit.

Regarding regions, women residing in city administrations (Addis Ababa and Dire Dawa) had 3.13 (AOR = 3.13; 95% CI: 1.77–5.55) times higher odds of institutional delivery as compared to women residing in pastoral regions. Women from communities with high media exposure had 2.01(AOR = 2.01; 95% CI: 1.44–2.79) times higher odds of institutional delivery as compared to women from a community with low media exposure. Besides, urban women had 4.70 (AOR = 4.70; 95% CI: 2.70–8.01) times higher odds of having institutional delivery as compared to rural residents (Table 5).

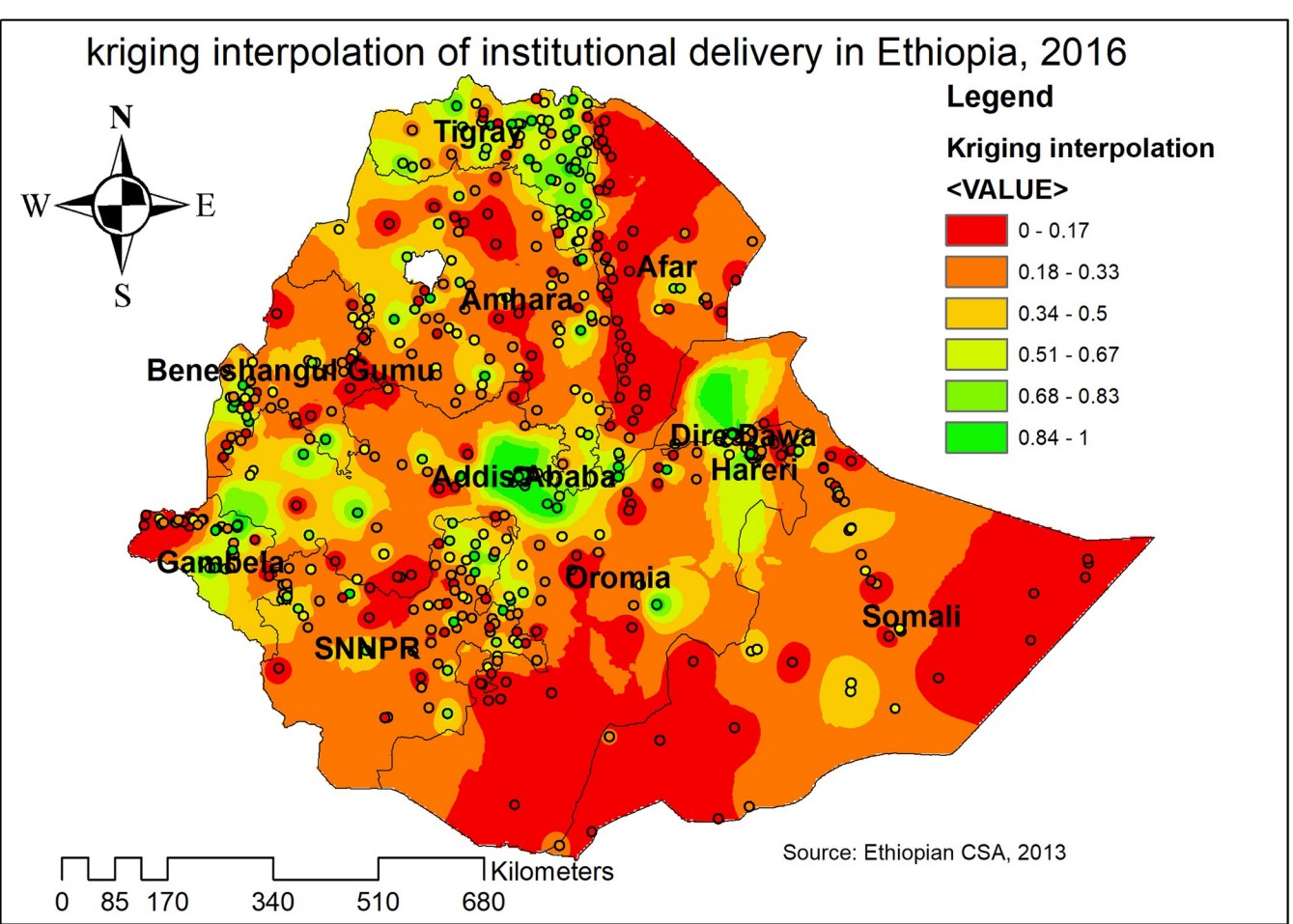

**Fig 5. Kriging interpolation of institutional delivery in Ethiopia, 2016 (Source, CSA: 2013).**

## Discussion

The most effective intervention to prevent maternal mortality from the major causes of maternal death, such as bleeding, sepsis, eclampsia, and obstructed labor is institutional delivery [47]. The current study was aimed to investigate the individual and community level determinants, and spatial distribution of institutional delivery in Ethiopia based on the nationally representative EDHS data.

The prevalence of institutional delivery in Ethiopia was found to be 26.2% in this analysis. It was lower than the prevalence in Nepal [10] and Tanzania [1], it could be due to the differences in accessibility and availability of maternal health care services across countries as Nepal and Tanzania have comparatively better socio-economic status compared to Ethiopia. While it was higher than previous studies reported in Ethiopia [30], and Bangladesh [50]. This may be attributed to the strengthened political commitment of Ethiopia for enhancing maternal health care services availability and accessibility by establishing health extension programs, expansion of health facilities, increased qualified health professionals, and improved quality of service [48–50].

The spatial analysis found that the spatial distribution of institutional delivery across the country was substantially varied. In the Harari, southern Oromia, and most parts of the Somali

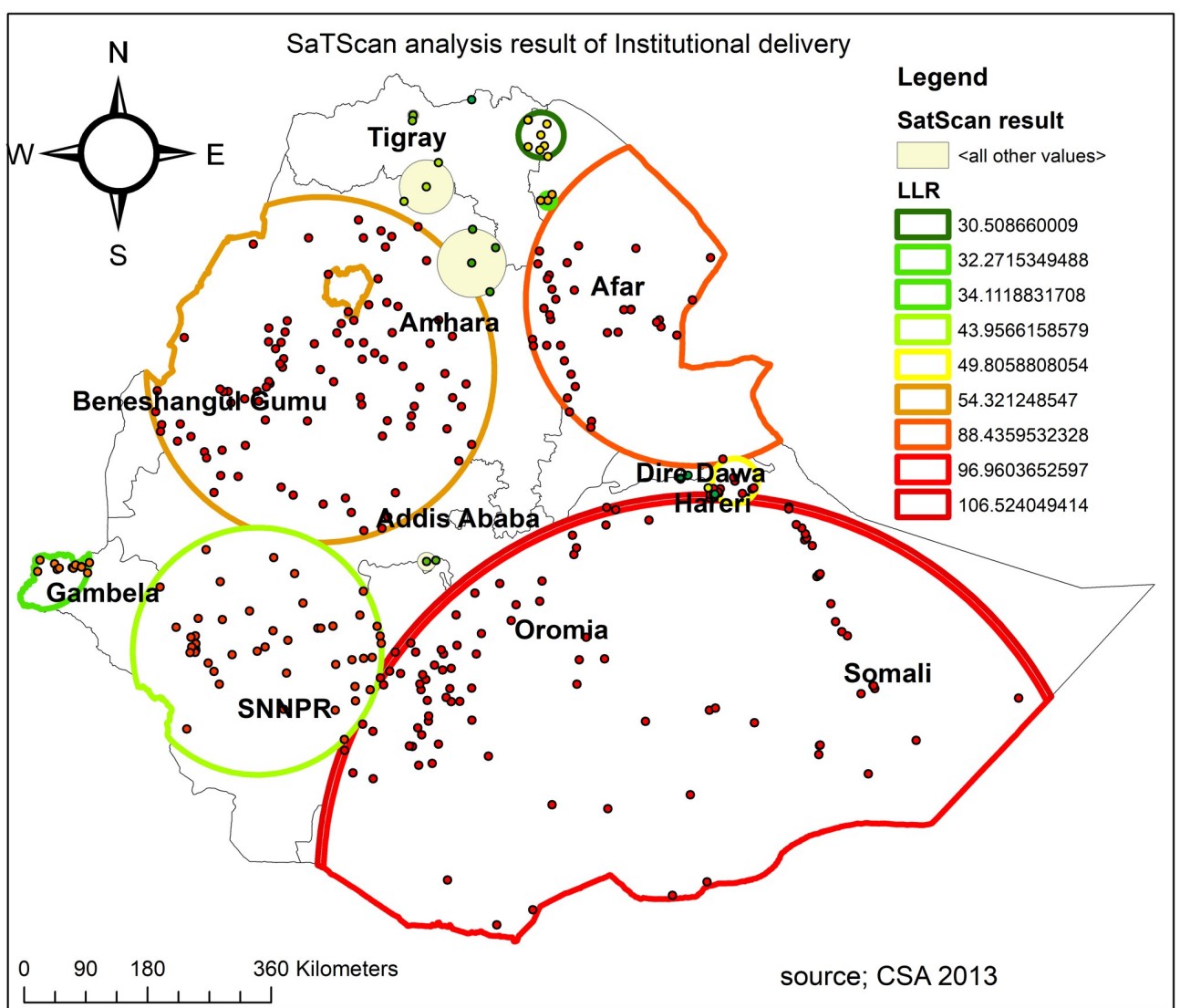

**Fig 6. SaTScan analysis of hotspot areas of poor institutional delivery (home delivery) in Ethiopia, 2016 (Source, CSA: 2013).**

regions, significant hotspot areas with a low prevalence of institutional delivery (high home delivery) were established. The possible explanation might be due to the disparity in the unavailability of maternal health services, and the inaccessibility of infrastructure such as road for transportation in the border regions of those regions [51]. Besides, these areas are more pastoral areas where individuals have no permanent residents, as a result, compared to other areas, comparatively health facilities are not open and accessible [52]. This finding suggests that public health planners and programmers should design effective public health interventions to enhance institutional delivery in these significant hotspot areas where institutional delivery was low.

In the multilevel logistic regression analysis; preceding birth interval, the number of ANC visits, wealth status, residence, community media exposure, region, and maternal education were significantly associated with institutional delivery. Among individual-level factors, maternal education was found to be a significant predictor of institutional delivery. Women who

**Table 3. SaTScan analysis result of home delivery.**

| Cluster | Enumeration area(cluster)identified | Coordinate/radius | Population | Case | RR | LLR | p-value |
|---|---|---|---|---|---|---|---|
| 1 (104) | 520, 208, 556, 394, 278, 164, 187, 480, 377, 318, 7, 358, 85, 138, 82, 289, 492, 286, 146, 422, 92, 543, 472, 490, 601, 452, 171, 198, 34, 95, 398, 316, 497, 518, 405, 21, 468, 313, 232, 600, 576, 445, 182, 26, 521, 574, 588, 562, 32, 123, 553, 458, 634, 365, 619, 213, 12, 319, 589, 215, 216, 308, 391, 408, 50, 148, 214, 578, 529, 251, 573, 245, 77, 239, 524, 503, 522, 116, 372, 22, 342, 347, 438, 609, 476, 122, 505, 20, 420, 162, 568, 412, 277, 86, 53, 513, 454, 373, 180, 580, 68, 506, 450, 501 | (4.180558 N, 42.052871 E) / 567.56 km | 2359 | 1877 | 1.24 | 106.5 | <0.0001 |
| 2 (91) | 520, 208, 556, 394, 278, 164, 187, 480, 377, 318, 7, 358, 85, 138, 82, 289, 492, 286, 146, 422, 92, 543, 472, 490, 601, 452, 171, 198, 34, 95, 398, 316, 497, 518, 405, 21, 468, 313, 232, 600, 576, 445, 182, 26, 521, 574, 588, 562, 32, 123, 553, 458, 634, 365, 619, 213, 12, 319, 589, 215, 216, 308, 391, 408, 50, 148, 214, 578, 529, 251, 573, 245, 77, 239, 524, 503, 522, 116, 372, 22, 342, 347, 438, 609, 476, 122, 505, 20, 420, 162, 568 | (4.180558 N, 42.052871 E) / 558.39 km | 2035 | 1630 | 1.24 | 96.96 | <0.001 |
| 3 (36) | 4, 632, 75, 596, 440, 366, 178, 499, 205, 427, 334, 570, 348, 599, 544, 389, 368, 241, 55, 547, 191, 571, 344, 276, 332, 189, 254, 37, 249, 620, 488, 307, 135, 611, 345, 283 | (11.845228 N, 41.915793 E) / 242.50 km | 697 | 617 | 1.34 | 88.44 | <0.001 |
| 4 (91) | 109, 3, 361, 498, 515, 382, 516, 615, 429, 541, 375, 548, 431, 167, 602, 246, 533, 494, 474, 403, 559, 386, 259, 73, 24, 169, 415, 36, 150, 184, 456, 158, 183, 120, 531, 218, 137, 512, 244, 292, 364, 132, 482, 206, 35, 229, 350, 320, 163, 38, 161, 176, 88, 627, 294, 399, 279, 10, 280, 70, 545, 640, 327, 256, 510, 124, 52, 621, 517, 65, 349, 267, 460, 234, 569, 152, 312, 199, 638, 335, 485, 304, 457, 423, 118, 209, 572, 324, 23, 563, 628 | (10.934452 N, 36.945496 E) / 252.94 km | 1434 | 1135 | 1.34 | 54.32 | <0.01 |
| 5 (9) | 566, 1, 186, 622, 8, 436, 210, 212, 419 | (9.455401 N, 42.455144 E) / 33.24 km | 240 | 225 | 1.40 | 49.81 | <0.01 |
| 6 (49) | 207, 154, 477, 489, 76, 338, 586, 177, 325, 437, 376, 168, 552, 459, 243, 299, 465, 371, 554, 470, 486, 526, 432, 197, 119, 46, 447, 555, 306, 227, 326, 62, 113, 411, 406, 141, 337, 126, 502, 434, 558, 565, 448, 180, 142, 331, 41, 360, 450 | (7.220845 N, 36.133859 E) / 180.87 km | 903 | 731 | 1.22 | 43.96 | 0.06 |
| 7 (12) | 266, 618, 309, 435, 536, 370, 507, 592, 104, 260, 233, 69 | (8.389747 N, 33.258557 E) / 71.61 km | 203 | 186 | 1.37 | 34.11 | 0.08 |
| 8 (3) | 130, 511, 172 | (13.169308 N, 39.987117 E) / 10.69 km | 82 | 82 | 1.49 | 32.27 | 0.1 |

attained primary and secondary education had a higher likelihood of institutional delivery than women who didn't attain formal education. It is consistent with previous study findings [10, 53–57]. It might be because education is the key to adapting positive behaviors like utilizing maternal health care services and educated mothers might be well informed about the benefits of institutional delivery [58]. Furthermore, maternal education could lead to the corresponding improvement in the mothers' health-seeking behavior compared to un-educated women.

The odds of institutional delivery among women who had ANC visits during pregnancy were higher than those who didn't have ANC visits. It was consistent with studies reported in Ethiopia [59, 60], and Bangladesh [56]. This is due to the assumption that ANC visits during pregnancy may increase the awareness of women about the risks of pregnancy and childbirth, as well as helping the mother to have an effective birth preparedness plan, which may increase the chance of their delivery at health facilities [61]. Besides, health education, counseling, and

**Table 4. Random effect analysis result.**

| Parameter | Null model | Model1 | Model2 | Model3 |
|---|---|---|---|---|
| Community level variance (SE) | 4.41 (0.47) | 1.37 (0.16) | 1.65 (0.19) | 1.21(0.153) |
| Log likelihood | -4737.52 | -3015.16 | -4491.17 | -2952.70 |
| Deviance | 9475.04 | 6,030.32 | 8982.34 | 5905.40 |
| MOR | 7.01 [6.02, 9.17] | 3.05[2.70, 3.49] | 3.39[2.97, 3.92] | 2.84 [2.52, 3.27] |
| PCV | Ref | 0.69 | 0.62 | 0.73 |
| ICC | 0.57 | 0.29 | 0.33 | 0.27 |

**Table 5. Multivariable multilevel logistic regression analysis of individual and community level determinants of institutional delivery in Ethiopia, 2016.**

| Variable | Null model | Model 1 | Model 2 | Model 3 |
|---|---|---|---|---|
| **Individual level factors** | | | | |
| **Parity** | | | | |
| 1 | | 1 | | 1 |
| 2–4 | | 0.49 [0.39, 0.62]** | | 0.48 [0.34, 0.68]** |
| >4 | | 0.46 [0.34, 0.61]** | | 0.48 [0.32, 0.74]** |
| **Women age (in years)** | | | | |
| <20 | | 1 | | 1 |
| 20–34 | | 0.87 [0.62, 1.21] | | 0.81 [0.52, 1.26] |
| ≥35 | | 0.88 [0.60, 1.30] | | 0.78 [0.45, 1.35] |
| **Preceding birth interval (in months)** | | | | |
| < 24 | | 1 | | 1 |
| 24–47 | | 1.25 [0.99, 1.58] | | 1.26 [0.88, 1.80] |
| ≥ 48 | | 1.52 [1.18, 1.95]** | | 1.51 [1.03, 2.20]* |
| **Maternal education** | | | | |
| No education | | 1 | | 1 |
| Primary | | 1.52 [1.28, 1.80]** | | 1.47 [1.16, 1.87]** |
| Secondary and higher | | 3.94 [2.82, 5.51]** | | 3.44[2.19, 5.42]** |
| **Husband education** | | | | |
| No education | | 1 | | 1 |
| Primary | | 1.07 [0.90, 1.27] | | 1.06 [0.81, 1.40] |
| Secondary and above | | 1.41 [1.13, 1.76]** | | 1.36 [0.99, 1.86] |
| **Number of ANC visit** | | | | |
| None | | 1 | | 1 |
| 1–3 | | 4.04 [3.31, 4.94]** | | 3.88 [2.77, 5.43]** |
| ≥ 4 | | 7.15 [5.85, 8.73]** | | 6.53 [4.69, 9.10]** |
| **Wealth status** | | | | |
| Poorest | | 1 | | 1 |
| Poorer | | 1.67 [1.31, 2.13]** | | 1.59 [1.10,2.30]* |
| Middle | | 1.54 [1.20, 1.97]** | | 1.44 [0.99, 2.08] |
| Richer | | 1.64 [1.26, 2.13]** | | 1.46 [0.99, 2.16] |
| Richest | | 5.20 [3.82, 7.09]** | | 2.44 [1.54, 3.87]** |
| **Sex of household head** | | | | |
| Female | | 1.06 [0.86, 1.30] | | 1.01 [0.74, 1.38] |
| Male | | 1 | | 1 |
| **Media exposure** | | | | |
| No | | 1 | | 1 |
| Yes | | 1.12 [0.95, 1.32] | | 1.00 [0.78, 1.29] |
| **Covered by health insurance** | | | | |
| No | | 1 | | 1 |
| Yes | | 1.58 [1.12, 2.24]** | | 1.59 [0.93, 2.74] |
| **Multiple gestation** | | | | |
| Single | | 1 | | 1 |
| Twin | | 2.45 [1.49, 4.01]** | | 2.35 [0.96, 5.76] |
| **Community level factors** | | | | |
| **Distance to health facility** | | | | |
| Not a big concern | | | 1 | 1 |
| Big concern | | | 1.34[1.10, 1.63]** | 1.20 [0.96, 1.51] |

*(Continued)*

**Table 5.** (Continued)

| Variable | Null model | Model 1 | Model 2 | Model 3 |
|---|---|---|---|---|
| **Residence** | | | | |
| Rural | | | 1 | 1 |
| Urban | | | 15.26[9.89, 23.56] | 4.70 [2.76, 8.01]** |
| **Community media exposure** | | | | |
| Lower | | | 1 | 1 |
| Higher | | | 2.91 [2.11, 4.01] | 2.01 [1.44, 2.79]** |
| **Region** | | | | |
| Pastoral | | | 1 | 1 |
| Semi pastoral | | | 3.17 [1.97, 5.10] | 1.37 [0.80, 2.33] |
| City administration | | | 12.91[7.6321.83] | 3.13 [1.77, 5.55]** |
| Agrarian | | | 3.33 [2.26, 4.92] | 1.39 [0.89, 2.17] |
| Constant | 0.49[0.40,0.60] | 0.08 [0.05, 0.12] | 0.04 [0.03, 0.06] | 0.04 [0.02, 0.08] |

Note;

* = p-value<0.05,

** = p-value<0.01.

treatment services offered by the health professional during ANC visits can result in women's behavioral changes and increased perceived benefits of seeking institutional delivery services [59].

Consistent with previous studies [10, 53, 56, 57], this study revealed that household wealth status was a significant predictor of institutional delivery. The likelihood of having institutional delivery was higher among mothers in the richest household wealth index than the poorest. These may be because better economic status may increase healthcare-seeking behavior and autonomy of healthcare decision-making as they are capable of paying the required medical and transport costs. [62]. While maternity and ambulance services are free in Ethiopia, it is still well known that drug and transportation services are still out of pocket charge, as many of the drugs are not accessible in public health facilities and there is a small number of ambulances.

In our study, multiparty was significantly associated with decreased odds of institutional delivery compared to primiparous women and this was consistent with previous study findings [10, 53]. This may be because primiparous women are afraid that they are more vulnerable to complications during childbirth and seek early maternity care services, which makes them more likely to give birth at the delivery of health facilities [63]. Also, multiparous women often choose to give birth at home for the gain of privacy and believe they will not face problems and are familiar with childbirth [64]. Furthermore, institutional delivery seeking behavior is affected by the delivery service satisfaction of the preceding pregnancies.

Among the community-level factors, women from the community with high media exposure had higher odds of institutional delivery. This was supported by prior studies [2, 43, 53, 56]. The possible reason is that health information may enhance health-seeking habits through different electronic and print media, as information about what service is available, where and when to get the services, as well as the advantages and risks of accessing specific services, can be transmitted through such media [65]. The study revealed that the place of residence was found to be a significant predictor of institutional delivery. Women living in rural areas had a higher likelihood of having institutional delivery than rural residents. It was consistent with studies in Ethiopia [66, 67], Bangladesh [53, 56], and Nepal [10]. The possible explanation

could be due to urban women had better access to maternal health care services and alternative service provisions like the use of private sectors, and get access to transportation at a reasonable cost and time as compared to rural women [68]. Furthermore, urban residents are closer to information about the health benefits of institutional delivery. Besides, women in city administrations (Addis Ababa and Dire-Dawa) had higher odds of institutional delivery as compared to those from pastoral regions. The consistent result has been reported in Ethiopia [67, 69]. The possible justification is health facilities are easily accessible and highly concentrated in Addis Ababa and Dire-Dawa. But women in pastoral regions have poor access to education and are not permanent residents and because of these in these areas, there is limited availability and accessibility of maternal health services such as institutional delivery.

## Strength and limitations of the study

This study had strengths. First, the study was based on weighted data to make it representativeness at national and regional levels: therefore, it can be generalized to all women who gave birth during the study period. Besides, the study was based on an advanced (appropriate) model, by taking into account the clustering effect, to get reliable standard error and estimate. Moreover, the use of GIS and SaTScan statistical tests helps to detect similar and statistically significant hotspot areas of institutional delivery and design effective public health programs. But this study was not without limitations. The SaTScan detect only circular clusters, irregularly shaped clusters were not identified. Besides, the GPS data (Latitude and Longitude) taken at enumeration area were displaced to 5 Km in urban areas and 10 Km in Rural areas for the privacy issue, this could bias our spatial result. Furthermore, the EDHS survey did not incorporate clinically confirmed data; rather, it relied on mothers or caregivers reports and might have the possibility of social desirability and recall bias (27). Furthermore, due to the cross-sectional nature of the data, the temporal relationship can't be established.

## Conclusions

Institutional delivery utilization in Ethiopia was very low. The spatial distribution of institutional delivery was significantly varied in Ethiopia. The significant hotspot areas with a low prevalence of institutional delivery (high home delivery) were detected in the Harari, south Oromia, and most parts of Somali regions. Parity, preceding birth interval, maternal education, number of ANC visits, wealth status, residence, and region were found to be significantly associated with institutional delivery. Therefore, public health interventions targeting significant hotspot areas (areas with a low prevalence of institutional delivery) is essential to enhance institutional delivery and reduce maternal and newborn mortality. Besides, governmental and non-governmental organizations should scale up maternal health programs to rural and poorest women. For future researchers, it is good to incorporate maternal and community knowledge, attitude, and behavior towards maternal health care service utilization by using a mixed approach (qualitative and quantitative studies) to have a deeper understanding of the factors that impede them to give birth at the health facility.

## Acknowledgments

Our thanks go to MEASURE DHS Program which permitted us to use the EDHS data set.

## Author Contributions

**Conceptualization:** Getayeneh Antehunegn Tesema, Tesfaye Hambisa Mekonnen.

**Data curation:** Getayeneh Antehunegn Tesema, Achamyeleh Birhanu Teshale.

**Formal analysis:** Getayeneh Antehunegn Tesema, Tesfaye Hambisa Mekonnen, Achamyeleh Birhanu Teshale.

**Methodology:** Getayeneh Antehunegn Tesema, Tesfaye Hambisa Mekonnen, Achamyeleh Birhanu Teshale.

**Software:** Getayeneh Antehunegn Tesema, Tesfaye Hambisa Mekonnen, Achamyeleh Birhanu Teshale.

**Validation:** Getayeneh Antehunegn Tesema.

**Visualization:** Getayeneh Antehunegn Tesema, Tesfaye Hambisa Mekonnen, Achamyeleh Birhanu Teshale.

**Writing – original draft:** Getayeneh Antehunegn Tesema, Tesfaye Hambisa Mekonnen, Achamyeleh Birhanu Teshale.

**Writing – review & editing:** Getayeneh Antehunegn Tesema, Tesfaye Hambisa Mekonnen, Achamyeleh Birhanu Teshale.

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
