## [Decision Letter · Decision Letter 0]

12 Jun 2020

PONE-D-19-33251

Individual and community-level determinants and spatial distribution of institutional delivery in Ethiopia, 2016: Spatial and multilevel analysis

PLOS ONE

Dear Mr Tesema,

Thank you for submitting your manuscript to PLOS ONE. After careful consideration, we feel that it has merit but does not fully meet PLOS ONE’s publication criteria as it currently stands. Therefore, we invite you to submit a revised version of the manuscript that addresses the points raised during the review process.

We would appreciate receiving your revised manuscript by Jun 26 2020 11:59PM. To enhance the reproducibility of your results, we recommend that if applicable you deposit your laboratory protocols in protocols.io, where a protocol can be assigned its own identifier (DOI) such that it can be cited independently in the future. For instructions see: http://journals.plos.org/plosone/s/submission-guidelines#loc-laboratory-protocols

We look forward to receiving your revised manuscript.

Kind regards,

Professor Khaled Khatab, Ph.D.

Academic Editor

PLOS ONE

Additional Editor Comments (if provided):

1) Some similar studies have discussed this issue in the same country before. See, for example, Mekonnen, Z.A., Lerebo, W.T., Gebrehiwot, T.G. et al. Multilevel analysis of individual and community-level factors associated with institutional delivery in Ethiopia. BMC Res Notes 8, 376 (2015). https://doi.org/10.1186/s13104-015-1343-1; Mezmur M, Navaneetham K, Letamo G, Bariagaber H (2017) Individual, household and contextual factors associated with skilled delivery care in Ethiopia: Evidence from Ethiopian demographic and health surveys. PLoS ONE 12(9): e0184688. https://doi.org/10.1371/journal.pone.0184688.

1.a) So what this study added to the current knowledge?

1.b) A further comparisons with the above studies need to be included in the discussion section.

1.c) Also, the literature review needs to be more robust and to include all the similar that tackled this issue in Ethiopia.

2) It was not mentioned why the manuscript has focused on this only or this period only? Was there any inclusion/exclusion criteria considered?

4) Neither strength nor the limitations of this study were mentioned?

5) The future work plan was not mentioned and how we plan to overcome the limitations of this study in the future.

2. We note that Figures 1 and 4-6 in your submission contain map images which may be copyrighted. All PLOS content is published under the Creative Commons Attribution License (CC BY 4.0), which means that the manuscript, images, and Supporting Information files will be freely available online, and any third party is permitted to access, download, copy, distribute, and use these materials in any way, even commercially, with proper attribution. For these reasons, we cannot publish previously copyrighted maps or satellite images created using proprietary data, such as Google software (Google Maps, Street View, and Earth). For more information, see our copyright guidelines: http://journals.plos.org/plosone/s/licenses-and-copyright.

1.    You may seek permission from the original copyright holder of Figures 1 and 4-6 to publish the content specifically under the CC BY 4.0 license. 

Reviewers' comments:

Reviewer's Responses to Questions

**Comments to the Author**

1. Is the manuscript technically sound, and do the data support the conclusions?

Reviewer #1: Yes

Reviewer #2: Partly

2. Has the statistical analysis been performed appropriately and rigorously? 

Reviewer #1: Yes

Reviewer #2: No

3. Have the authors made all data underlying the findings in their manuscript fully available?

Reviewer #1: Yes

Reviewer #2: Yes

4. Is the manuscript presented in an intelligible fashion and written in standard English?

Reviewer #1: Yes

Reviewer #2: No

5. Review Comments to the Author

Reviewer #1: Summary:

The research focus though not new, it has helped to fill some of the identified gaps in the existing direction of the study.

The authors have demonstrated good understanding of the subject and had written the manuscript professionally and intelligently. Efforts put into the writing the manuscripts and the analysis have shown clearly that the authors are skilful and experienced researchers. The methodology adopted in the data analysis fits appropriately to the nature of the data and in fulfilment of the aim/objectives of the study. However, clearer research question(s) need to be established. The authors would need to consult experienced language editor to enhance the flow of the manuscripts.

Some minor errors/concerns need to be addressed. I have chronicle some the concerns I feel the authors need to address to enhance the quality of the paper and to meet the required standard of the intended journal (see the attached document).

Reviewer #2: Overview

This paper presents an interesting approach to understanding variation in institutional delivery in Ethiopia using both multilevel modeling and GIS. While the modeling strategy is interesting, the paper would be improved if the authors spent more time discussing the two approaches (the multilevel modeling and the GIS approach) in tandem and how they build on one another. More synthesis on the use of the added value and joint examination of the results would be interesting and would strengthen the paper.

General comments:

• The authors will need to carefully proof read the manuscript. In some places, it is difficult to follow because of grammatical errors. It may help to have an outside person edit the document for grammar. I’ve pulled a few examples here: page 1 line 63: “For example, the estimated 130,000 maternal deaths happened in 2017 in those countries [3].” Page 1 lines 71-72, “plentiful numbers of women in developing countries give birth at homes.” Page 2 line 90: “So far, Ethiopia has made a lot to curb maternal...”

• Page 1 line 67: I think it would be more accurate to say that strengthening facility delivery or skilled attendance would “reduce” not “alleviate” the burden of preventable maternal death.

• Page 2, line 96: is the low facility attendance due to limited health facilities, bad infrastructure, etc?

Methods

• There is a lot of background information provided in the “study design, setting and period” section. The authors may want to consider integrating some of the background about Ethiopia into the background section, as it is somewhat difficult to follow in the methods section.

• The authors should indicate that they are using DHS data and describe it from the beginning of the methods section when they first discuss the survey.

• Line 151: missing closed parentheses

• The authors may want to consider including the general equation that they use. It is somewhat confusing for the authors to describe the outcome variable as Yi given that multilevel models usually require a nested structure, so that individual i is usually nested in community j and so forth. Also, how is region a community level variable? Is that not a third level of the model? Or, are the authors including dummies for region as fixed effects?

• The authors should more clearly specify how they defined the level 2 community variables. As it stands, it is unclear how the authors measured community level media exposure. It would also be helpful for them to cite other papers that have used EA in DHS as a proxy for community.

• The calculation of the PCV in binary multilevel models is not straightforward due to the level-1 variation of the binary model. Can the authors describe in full what approach they used?

• The authors may want to consider condensing some of the material and reorganizing some of the subheadings in the methods section. For example, the data collection procedure section could be combined with the description of the survey. As of now, some of the information is duplicative and the section is somewhat difficult to follow.

• What statistical program did the authors use and using what estimation technique?

Results

• Does the fact that DHS throws the spatial coordinates of GPS influence to ensure that the exact location of the clusters is not revealed influence the results in any way? In particular, page 11 (the section beginning at line 292)?

• The interpretation of the odds ratio from the null model is confusing (line 305). I assume that 7.01 refers to the intercept? The authors may want to ensure that their interpretation is correct, and if so, explain it in a way that is more clear.

• The authors state that “About 73 percent of the variability in institutional delivery was explained by the full model.” Is this the model overall? What about the different levels?

• In the methods section, the authors describe “region” as a community-level variable. Does region refer to urban/rural as indicated in the results section?

• The authors may want to discuss community level variables separately from individual level variables as it is somewhat confusing to understand the results.

Discussion

• An important piece missing from the discussion is a synthesis of the results and a discussion of why context matters. The authors do a good job of summarizing the results and then discussing them in light of other literature, but it would be interesting for the authors to focus on understanding the combined results from both analytical approaches to deepen the understanding of the role of community context in Ethiopia.

• The authors do not discuss the results of the very interesting GIS analysis in the discussion.

6. PLOS authors have the option to publish the peer review history of their article (what does this mean?). If published, this will include your full peer review and any attached files.

Reviewer #1: No

Reviewer #2: No

---

## [Author Response · Author response to Decision Letter 0]

31 Jul 2020

PLOS ONE

Point by point response for editors/reviewers comments 

Manuscript title: Individual and community-level determinants and spatial distribution of institutional delivery in Ethiopia, 2016: Spatial and multilevel analysis

Manuscript ID: PONE-D-19-33251

Dear editor/reviewer. 

Dear all,

We would like to thank you for this constructive, building, and improvable comments on this manuscript that would improve the substance and content of the manuscript. We considered each comment and reviewers on the manuscript thoroughly. Our point-by-point responses for each comment and questions are described in detail on the following pages.

Response to editors comments

1. Some similar studies have discussed this issue in the same country before. See, for example, Mekonnen, Z.A., Lerebo, W.T., Gebrehiwot, T.G. et al. Multilevel analysis of individual and community-level factors associated with institutional delivery in Ethiopia. BMC Res Notes 8, 376 (2015). https://doi.org/10.1186/s13104-015-1343-1; Mezmur M, Navaneetham K, Letamo G, Bariagaber H (2017) Individual, household and contextual factors associated with skilled delivery care in Ethiopia: Evidence from Ethiopian demographic and health surveys. PLoS ONE 12(9): e0184688. https://doi.org/10.1371/journal.pone.0184688.

a) So what this study added to the current knowledge?

Authors’ response: Thank you editor for the concerns. These two previous studies were conducted based on EDHS 2005 and 2011 data to investigate the individual and community level determinants of institutional delivery using multilevel analysis. But both studies are failed to capture the spatial distribution of institutional delivery using ArcGIS and SaTScan analysis to identify the significant hotspot areas where institutional delivery was low even if studies done on the prevalence of institutional delivery in different parts of Ethiopia revealed that the prevalence has been varied across the country. Besides, in this study, the data were not weighted even if the DHS statistician recommended using weighted data based on strata, PSU, and weighting variables to restore the representativeness as well as to get reliable standard error and estimate. Therefore, we investigated the individual and community level determinants and spatial distribution of institutional delivery in Ethiopia based on the most recent EDHS data. Thus, exploring the spatial distribution of institutional delivery is important to identify significant primary and secondary hotspot areas where institutional delivery is low, this could help to design targeted public health interventions to the identified hotspot areas to enhance health facility delivery to reduce preventable maternal and newborn mortality. Furthermore, hence the data we used were weighted the estimates are reliable. (see the Backroung section, line 95-106, page 6)

b) A further comparisons with the above studies need to be included in the discussion section.

Authors’ response: Thank you editor for the comments. We had incorporated it. (see the revised document)

c) Also, the literature review needs to be more robust and to include all the similar that tackled this issue in Ethiopia

Authors’ response: Thank you, editor. We had incorporated previous study findings conducted on institutional delivery by extensively searching literature. (See the revised manuscript)

2. It was not mentioned why the manuscript has focused on this only or this period only? Was there any inclusion/exclusion criteria considered?

Authors’ response: Thank you editor for the concerns. We have used the EDHS 2016 data for this study since this Survey is the most recent in Ethiopia. For this study, we excluded the respondents where the outcome variable (place of delivery) was missed and for the spatial analysis, we excluded those women in the Enumeration Areas (EAs) with zero latitudes and longitude.

3. Neither strength nor the limitations of this study were mentioned?

Authors’ response: Thank you editor, we have included strength and limitations of the study in the revised manuscript. (See the Strengths and limitation section, line 427-438, page 21)

4. The future work plan was not mentioned and how we plan to overcome the limitations of this study in the future.

Authors’ response: Thank you editor, we have incorporated in the revised manuscript. (See the revised manuscript, Conclusion section, line 448-452, page 22)

5. We note that Figures 1 and 4-6 in your submission contain map images which may be copyrighted. All PLOS content is published under the Creative Commons Attribution License (CC BY 4.0), which means that the manuscript, images, and Supporting Information files will be freely available online, and any third party is permitted to access, download, copy, distribute, and use these materials in any way, even commercially, with proper attribution. For these reasons, we cannot publish previously copyrighted maps or satellite images created using proprietary data, such as Google software (Google Maps, Street View, and Earth). For more information, see our copyright guidelines: http://journals.plos.org/plosone/s/licenses-and-copyright.

Authors’ response: Thank you editor for the concern. The map is not copyrighted rather we have done using ArcGIS and SaTScan software based on the shapefile of Ethiopia received from Ethiopian Central Statistical Agency (CSA) by explaining the purpose of the study and GPS data (longitude and latitude) from measure DHS program by explaining the objective of the study through online requesting and allow us to access the shapefile and GPS data. Now we cite the source of the shapefile since it is needed to explore the spatial distribution of institutional delivery. Therefore, the maps presented in our study are not copyrighted rather it was our spatial analysis result.

Response to reviewers comments

Reviewer#1

1. The research focus though not new, it has helped to fill some of the identified gaps in the existing direction of the study.

The authors have demonstrated good understanding of the subject and had written the manuscript professionally and intelligently. Efforts put into the writing the manuscripts and the analysis have shown clearly that the authors are skilful and experienced researchers. The methodology adopted in the data analysis fits appropriately to the nature of the data and in fulfilment of the aim/objectives of the study. However, clearer research question(s) need to be established. The authors would need to consult experienced language editor to enhance the flow of the manuscripts.

Authors’ response: Thank you, reviewer, for the comments. The research questions in our study were, 1) to identify the individual and community level factors associated with institutional delivery, 2) to explore the spatial distribution of institutional delivery, and to identify significant hotspot areas where institutional delivery utilization is low to design targeted public health interventions. We had extensively edited and modified the entire document. (see the revised manuscript)

2. Some minor errors/concerns need to be addressed. I have chronicle some the concerns I feel the authors need to address to enhance the quality of the paper and to meet the required standard of the intended journal

Authors’ response: We thank the reviewer for your great effort for the betterment of our work. We accepted your comments and modified accordingly. (see the revised manuscript)

Reviewer# 2

1. this paper presents an interesting approach to understanding variation in institutional delivery in Ethiopia using both multilevel modeling and GIS. While the modeling strategy is interesting, the paper would be improved if the authors spent more time discussing the two approaches (the multilevel modeling and the GIS approach) in tandem and how they build on one another. More synthesis on the use of the added value and joint examination of the results would be interesting and would strengthen the paper.

Authors’ response: Thank you, reviewer. We synthesize both the use of spatial and multilevel modeling and for further we cite references. (See the revised manuscript)

2. The authors will need to carefully proof read the manuscript. In some places, it is difficult to follow because of grammatical errors. It may help to have an outside person edit the document for grammar. I’ve pulled a few examples here: page 1 line 63: “For example, the estimated 130,000 maternal deaths happened in 2017 in those countries [3].” Page 1 lines 71-72, “plentiful numbers of women in developing countries give birth at homes.” Page 2 line 90: “So far, Ethiopia has made a lot to curb maternal...” • Page 1 line 67: I think it would be more accurate to say that strengthening facility delivery or skilled attendance would “reduce” not “alleviate” the burden of preventable maternal death. • Page 2, line 96: is the low facility attendance due to limited health facilities, bad infrastructure, etc?

Authors’ response: Thank you, reviewer, for the comments. we had extensively edited the whole document with the help of language experts at university of Gondar. (See the revised manuscript)

3. There is a lot of background information provided in the “study design, setting and period” section. The authors may want to consider integrating some of the background about Ethiopia into the background section, as it is somewhat difficult to follow in the methods section. The authors should indicate that they are using DHS data and describe it from the beginning of the methods section when they first discuss the survey. Line 151: missing closed parentheses

Authors’ response: Thank you, reviewer. we have modified it. (see the revised manuscript)

4. The authors may want to consider including the general equation that they use. It is somewhat confusing for the authors to describe the outcome variable as Yi given that multilevel models usually require a nested structure, so that individual i is usually nested in community j and so forth. Also, how is region a community level variable? Is that not a third level of the model? Or, are the authors including dummies for region as fixed effects?

Authors’ response: Thank you, reviewer. we incorporated the general equation of the multilevel model we fitted. We use Enumeration area/clusters as a random effect as level two since women within the selected enumeration area are more correlated than the women from different clusters. In EDHS except for region, distance to the health facility, and residence the other variables were collected at the individual level. We tried to consider the region as the third level of the model but the result was the same as two level multilevel model and besides, we didn't get any variable collected at the region level. We categorized the region into four groups as Pastoral, Semi pastoral, City administration, and Agrarian in the multilevel analysis and considered as a community level variable at level 2. 

5. The authors should more clearly specify how they defined the level 2 community variables. As it stands, it is unclear how the authors measured community level media exposure. It would also be helpful for them to cite other papers that have used EA in DHS as a proxy for community.

Authors’ response: Thank you, reviewer. We included how we generated community-level variables. In EDHS data except for region, residence, and distance to health facility these variables were collected at individual levels. The EDHS data has hierarchical nature means women are nested within-cluster therefore we want to assess the individual and cluster level variables that affect institutional delivery. Then to assess whether community media exposure has a significant effect on institutional delivery we aggregated media exposure collected at the individual level to community level/cluster level. Then we categorized as high community media exposure and low community media exposure based on the national median value since it was not normally distributed. (See the revised manuscript)

6. The calculation of the PCV in binary multilevel models is not straightforward due to the level-1 variation of the binary model. Can the authors describe in full what approach they used?

Authors’ response: thank you, reviewer, for the comments. The proportional change in variance (PCV) measures the total variation attributed by individual-level factors and area-level factors in the multilevel model. PCV is used to show the total variability explained by the final model (model with individual and community level variable simultaneously) relative to the null model, it is like the coefficient of determination (R2) in the linear regression model. It is interpreted as the total variability of institutional delivery explained by the final model. As you know in the multilevel binary logistic regression model the level-1 variation is constant (π2/3) unlike the linear regression since there are no residuals. Therefore, we calculated by substracted the variance of institutional delivery in the final model from the variance in the null mode divided by the variance in the null model.

7. The authors may want to consider condensing some of the material and reorganizing some of the subheadings in the methods section. For example, the data collection procedure section could be combined with the description of the survey. As of now, some of the information is duplicative and the section is somewhat difficult to follow.

Authors’ response: Thank you, reviewer, we organized to combined related subheadings together. (See the revised manuscript)

8. What statistical program did the authors use and using what estimation technique?

Authors’ response: Thank you reviewer. For analysis we used melogit STATA command (melogit institutional_delivery i.ANC i.region i.parity i.residence i.communty_media_exposure i.maternal_education i.materna_age i.husband_education i.birth_interval i.wealth_status [pw=wgt] || v001:, or), using maximum likelihood estimation technique. To calculate the MOR and ICC we install the xtmrho stata command and run after running the full model.

9. Results

• Does the fact that DHS throws the spatial coordinates of GPS influence to ensure that the exact location of the clusters is not revealed influence the results in any way? In particular, page 11 (the section beginning at line 292)?

Authors’ response: thank you, reviewer, for the comments. In EDHS as GPS data (latitude and longitude) taken in the Enumeration area level were displaced up to 5km in Urban area, and up to 10 km in rural areas because of privacy issues, therefore, this could bias our findings and we acknowledge in the limitation sections of the study. (See the revised manuscript, limitation section, line 434-436, page 21)

10. The interpretation of the odds ratio from the null model is confusing (line 305). I assume that 7.01 refers to the intercept? The authors may want to ensure that their interpretation is correct, and if so, explain it in a way that is more clear.

Authors' response: Thank you, reviewer. This MOR result in the null model. MOR quantifies the variation between clusters (the second level variations) by comparing two women from two randomly chosen, different clusters. The result was MOR= 7.01, 95% CI: 6.02, 9.17 it indicates that if we randomly select two women from two different clusters. A woman from the cluster with a high likelihood of institutional had 7.01 times higher odds of having institutional delivery compared with women form clusters with lower institutional delivery. Therefore, this finding was not the intercept rather it is another method of assessing heterogeneity across clusters like ICC. 

11. The authors state that “About 73 percent of the variability in institutional delivery was explained by the full model.” Is this the model overall? What about the different levels?

Authors’ response: Thank you, reviewer, for the concern. PCV is used to measure the total variation in institutional delivery explained by the final model (a model with both individual and community-level factors simultaneously) in relative to the null model ( Model without independent variables).

Therefore, about 73% of the total variation in institutional delivery was explained by the overall model.

12. In the methods section, the authors describe “region” as a community-level variable. Does region refer to urban/rural as indicated in the results section?

Authors' response: Thank you, reviewer, for the comments. In Ethiopia, there are 9 regions and 2 city administrations. In our study, we categorized the region into 4 groups. 1, pastoralist region (Benishangul, Somali, Gambella, and Afar), Semipastorlaist (Oromia, SNNPR), Agrarian (Amhara and Tigray) and City administration (Addis Ababa, Dire Dawa, and Harari) based on literature. Rural and Urban were for the variable residence. 

13. The authors may want to discuss community level variables separately from individual level variables as it is somewhat confusing to understand the results.

Authors' response: Thank you, reviewer. we have discussed it separately, first, we discussed the individual variables significantly associated with institutional delivery. (See the Discussion section, line 347-425, page 17-21)

14. Discussion

• An important piece missing from the discussion is a synthesis of the results and a discussion of why context matters. The authors do a good job of summarizing the results and then discussing them in light of other literature, but it would be interesting for the authors to focus on understanding the combined results from both analytical approaches to deepen the understanding of the role of community context in Ethiopia.

Authors' response: Thank you, reviewer. we have written out in the revised manuscript.

15. • The authors do not discuss the results of the very interesting GIS analysis in the discussion.

Authors' response: Thank you, reviewer. we have discussed it. (See the Discussion section, line 361-370, page 18)

---

## [Decision Letter · Decision Letter 1]

28 Sep 2020

PONE-D-19-33251R1

Individual and community-level determinants, and spatial distribution of institutional delivery in Ethiopia, 2016: Spatial and multilevel analysis

PLOS ONE

Dear Mr Tesema,

Thank you for submitting your manuscript to PLOS ONE. After careful consideration, we feel that it has merit but does not fully meet PLOS ONE’s publication criteria as it currently stands. Therefore, we invite you to submit a revised version of the manuscript that addresses the points raised during the review process.

Please submit your revised manuscript by 7th of November 11:59PM. If you will need more time than this to complete your revisions, please reply to this message or contact the journal office at plosone@plos.org. Please include the following items when submitting your revised manuscript:

We look forward to receiving your revised manuscript.

Kind regards,

Prof Khaled Khatab, Ph.D.

Academic Editor

PLOS ONE

Reviewers' comments:

Reviewer's Responses to Questions

**Comments to the Author**

1. If the authors have adequately addressed your comments raised in a previous round of review and you feel that this manuscript is now acceptable for publication, you may indicate that here to bypass the “Comments to the Author” section, enter your conflict of interest statement in the “Confidential to Editor” section, and submit your "Accept" recommendation.

Reviewer #2: (No Response)

Reviewer #3: All comments have been addressed

2. Is the manuscript technically sound, and do the data support the conclusions?

Reviewer #2: Yes

Reviewer #3: Yes

3. Has the statistical analysis been performed appropriately and rigorously? 

Reviewer #2: I Don't Know

Reviewer #3: Yes

4. Have the authors made all data underlying the findings in their manuscript fully available?

Reviewer #2: Yes

Reviewer #3: Yes

5. Is the manuscript presented in an intelligible fashion and written in standard English?

Reviewer #2: No

Reviewer #3: Yes

6. Review Comments to the Author

Reviewer #2: The authors have done a good job in addressing many of my previous comments, but there still remain several issues for the authors to address.

General

There are still some grammatical issues that the authors should address prior to publication. In particular, there are still grammatical mistakes, misplaced words, and incorrect/inconsistent use of tense.

Methods

1. The organization of the methods section is still somewhat confusing. The multilevel equation specified by the authors appears under the outcome variable description when it seems like the discussion of the model would be more appropriate under the description of multilevel analysis that appears later in the methods section.

2. The authors describe their calculation of the PCV as being reliant on calculation the total variance. The authors should include how they calculate the total variance, given that the VPC is more difficult to calculate in binary-response models due to the different scale of the level 1 variance because of use of the link function.

3. Furthermore, the authors may want to discuss the implications of their choice of estimation technique (maximum likelihood) on the estimates of variance, given that maximum likelihood estimation tends to underestimate the variance at higher levels in multilevel binary models, and often MCMC estimation is used instead when the variance parameters are of substantive interest. The authors may want to consider adding this in the limitations section.

Results

4. The authors calculate the total variance explained by the addition of the covariates in the model, but they do not decompose the variance to examine the variance explained at the community level. This would be helpful, and again goes back to my previous comment asking the authors about how they calculated the total variation in the model.

Discussion

1. The discussion still requires some grammatical editing and certain parts are difficult to follow. In particular, paragraph 2.

Reviewer #3: Authors have addressed concerned earlier. Having reviewed the current version of the manuscript I can confidently confirmed that the article has now significantly met the set conditions for it to be published in this journal. Thus, I hereby recommend that the manuscript be accepted for publication. The only area I wish the authors should try and adjust is in the equations in lines 135-142. Using equation editor could help to fine-tune the notations/subscripts of the respective equation's parameters.

For record however, I commend the authors for the efforts channeled into writing the manuscript and I believe their scholastic contributions into the research community would receive wider acceptability.

7. PLOS authors have the option to publish the peer review history of their article (what does this mean?). If published, this will include your full peer review and any attached files.

Reviewer #2: No

Reviewer #3: No

---

## [Author Response · Author response to Decision Letter 1]

8 Oct 2020

Point by point response for editors/reviewers comments 

Manuscript title: Individual and community-level determinants, and spatial distribution of institutional delivery in Ethiopia, 2016: Spatial and multilevel analysis

Manuscript ID: PONE-D-19-33251R1

Dear editor/reviewer. 

Dear all,

We would like to thank you for this constructive, building, and improvable comments on this manuscript that would improve the substance and content of the manuscript. We considered each comment and clarification questions of editors and reviewers on the manuscript thoroughly. Our point-by-point responses for each comment and questions are described in detail on the following pages. Further, the details of changes were shown by track changes in the supplementary document attached.

Response to reviewers comment

Reviewer # 2

1. General

There are still some grammatical issues that the authors should address prior to publication. In particular, there are still grammatical mistakes, misplaced words, and incorrect/inconsistent use of tense.

Authors’ response: Thank you reviewer for the comments. We extensively edited the whole document for the grammatical error, typical errors, and sentence structure with the help of language experts at UOG. (See the revised manuscript)

2. Methods

2.1. The organization of the methods section is still somewhat confusing. The multilevel equation specified by the authors appears under the outcome variable description when it seems like the discussion of the model would be more appropriate under the description of multilevel analysis that appears later in the methods section.

Authors’ response: Thank you reviewer for raising the important comments. We accept your comments and we placed the equation in the multilevel analysis section. (See the Method section, line 191- 202, page 10)

2.2. . The authors describe their calculation of the PCV as being reliant on calculation the total variance. The authors should include how they calculate the total variance, given that the VPC is more difficult to calculate in binary-response models due to the different scale of the level 1 variance because of use of the link function.

Authors’ response: Thank you reviewer for the comments. As you stated, it is difficult to calculate VPC/ICC is more difficult since unlike the linear regression model the individual-level variance in logistic regression is assumed to follow the standard logistic distribution with mean 0 and variance of π2/3 (3.29). So, the individual level variance is constant that is π2/3, we add the cluster level variance with π2/3 to get the total variance in the null model and in the final model. Then we calculate PCV by using the formula (variance in the null model – variance in the final model)/variance in the null model.

2.3. Furthermore, the authors may want to discuss the implications of their choice of estimation technique (maximum likelihood) on the estimates of variance, given that maximum likelihood estimation tends to underestimate the variance at higher levels in multilevel binary models, and often MCMC estimation is used instead when the variance parameters are of substantive interest. The authors may want to consider adding this in the limitations section

Authors’ response: Thank you reviewer for the concerns. We used the maximum likelihood estimation technique to estimate the estimates of variance since we used the classical multilevel binary logistic regression and the sample is large. We plan to use MCMC to estimate the variance using MCMC but to do this it needs the posterior distribution (bayesian approach). And now we have checked by simulating the data and estimate the variance using MCMC but the value is the same as the MLE findings since the sample is adequate and the prevalence is 26.2.

2.4. Results, The authors calculate the total variance explained by the addition of the covariates in the model, but they do not decompose the variance to examine the variance explained at the community level. This would be helpful, and again goes back to my previous comment asking the authors about how they calculated the total variation in the model.

Authors’ response: Thank you reviewer for concerns. PCV is commonly done in multilevel analysis to examine by how much the full model explains the variations of institutional delivery as compared to the null model. And we calculate the PCV for three models (model1 model2, and model 3) in relative to the null model. Logically we need that the PCV value to be highest in the final model as the number of covariates incorporated is increased, based on these the final model was 73% indicates that the final model explains the variation in institutional delivery by 73%, it is like the coefficient of determination in the linear regression model. So, this 73% is explained by the mutual effects of the individual as well as community-level variables, and it is difficult to partition how much is explained by the individual and how much is by the community level variables. Just PCV is supportive, beyond that we have used deviance, MOR, ICC, and LR-test to check whether the model is improved or not. Just to know how much is attributable to the community level and how much is by community level we can get in model 1 and model 2 as we presented in the table.

3. Discussion

1. The discussion still requires some grammatical editing and certain parts are difficult to follow. In particular, paragraph 2.

Authors’ response: Thank you the reviewer for the comments. We extensively modify it. (See the revised manuscript)

Reviewer #3

1. The authors have addressed concerned earlier. Having reviewed the current version of the manuscript I can confidently confirmed that the article has now significantly met the set conditions for it to be published in this journal.

Thus, I hereby recommend that the manuscript be accepted for publication. The only area I wish the authors should try and adjust is in the equations in lines 135-142. Using equation editor could help to fine-tune the notations/subscripts of the respective equation's parameters

Authors’ response: Thank you reviewer for the comments. We accept the comment and we put in equation form. (See the revised manuscript)

---

## [Editor Report · Decision Letter 2]

30 Oct 2020

Individual and community-level determinants, and spatial distribution of institutional delivery in Ethiopia, 2016: Spatial and multilevel analysis

PONE-D-19-33251R2

 Dear Mr Tesema,

We’re pleased to inform you that your manuscript has been judged scientifically suitable for publication and will be formally accepted for publication once it meets all outstanding technical requirements.

Within one week, you’ll receive an e-mail detailing the required amendments. When these have been addressed, you’ll receive a formal acceptance letter, and your manuscript will be scheduled for publication.

An invoice for payment will follow shortly after the formal acceptance. To ensure an efficient process, please log into Editorial Manager at http://www.editorialmanager.com/pone/, click the 'Update My Information' link at the top of the page, and double-check that your user information is up-to-date. If you have any billing-related questions, please contact our Author Billing department directly at authorbilling@plos.org.

Kind regards,

Professor Khaled Khatab, Ph.D.

Academic Editor

PLOS ONE
---

## [Editor Report · Acceptance letter]

4 Nov 2020

PONE-D-19-33251R2 

Individual and community-level determinants, and spatial distribution of institutional delivery in Ethiopia, 2016: Spatial and multilevel analysis 

Dear Dr. Tesema:

I'm pleased to inform you that your manuscript has been deemed suitable for publication in PLOS ONE. Congratulations! Your manuscript is now with our production department. 

Kind regards, 

on behalf of

Professor Khaled Khatab 

Academic Editor

PLOS ONE